# RETRACTED: Reactive Mesoporous pH-Sensitive Amino-Functionalized Silica Nanoparticles for Efficient Removal of Coomassie Blue Dye

**DOI:** 10.3390/nano9121721

**Published:** 2019-12-02

**Authors:** Nourah I. Sabeela, Tahani M. Almutairi, Hamad A. Al-Lohedan, Abdelrahman O. Ezzat, Ayman M. Atta

**Affiliations:** 1Surfactants Research Chair, Chemistry Department, College of Science, King Saud University, Riyadh 11451, Saudi Arabia; noonah-37@hotmail.com (N.I.S.); hlohedan@ksu.edu.sa (H.A.A.-L.); ao_ezzat@yahoo.com (A.O.E.); 2Chemistry Department, College of Science, King Saud University, Riyadh 11451, Saudi Arabia; talmutarai@ksu.edu.sa

**Keywords:** nano-adsorbent, amino-functionalized silica, nanoparticles, mesoporous, Coomassie blue

## Abstract

In this work, new smart mesoporous amine-functionalized silica nanoparticles were prepared from hydrolyzing microgels based on *N*-isopropyl acrylamide-*co*-vinyltrimethoxysilane microgels with tetraethoxysilicate and 3-aminopropyltriethoxysilane by sol-gel method. The thermal stability and Fourier transform infrared were used to determine the amine contents of the silica nanoparticles. The pH sensitivity of the synthesized silica nanoparticles in their aqueous solutions was evaluated by using dynamic light scattering (DLS) and zeta potential measurements. The porosity of the amine-functionalized silica nanoparticles was evaluated from a transmittance electron microscope and Brunauer-Emmett-Teller (BET) plot. The results have positively recommended the pH-sensitive amine-functionalized silica nanoparticles as one of the effective nano-adsorbent to remove 313 mg·g^−1^ of CB-R250 water pollutant.

## 1. Introduction

Nano-adsorbents attracted great attention among many types of conventional adsorbents, ion exchange, and crosslinked polymeric and composite materials used for different environmental and biomedical applications [1,2,3]. Their higher surface area, uniform sizes, reactivity, and lower cost are goal to favor their use as adsorbent for inorganic and organic water pollutants [4,5,6,7]. Their lower chemical resistivity, aggregations into large clusters due to their higher reactivity, biological activity, and their ability to undergo rapid biodegradation when exposed to the biological system have limited their applications [8]. Silica is one of the most important inert porous and mesoporous nanomaterials that have large pore sizes and strong chemical stability, besides the presence of terminated silanol groups on their surfaces that facilitate their surface modifications to increase their applicability in the medical and environmental field [9,10,11,12]. The silanol groups can be modified for excellent selectivity toward specific pollutants [13]. The mesoporous silica fulfills most of the criteria to be used as adsorbents for large-molecule organic and inorganic pollutants [14,15,16]. Several studies reported the use of silica materials to remove inorganic pollutants as compared to their lower ability to remove the organic pollutants [12]. The lower reactivity of silica to remove organic water pollutants that limited their application in aqueous solutions is due to the formation of hydrogen bond between water and silanol functional groups on the silica surfaces that restricted their interaction with the organic water pollutants.

Many chemical modifications have been proposed to improve the efficiency of the silica-based materials either by nitrogen or thiol organic surface modifications to increase the electrostatic interactions between silica and organic water pollutants [17,18,19,20]. The amino-propyl-functionalized silica improved the efficiency of mesoporous silica to remove humic and tannic acids from water, with an adsorption capacity of 272.4 mg·g^−1^ [17]. It was also reported that the functionalization of silica with more than one functional group improved their adsorption performance toward organic pollutants [18,19]. The monodisperse grafts of amino-terminated polyacrylic acid silica nanoparticles has improved the adsorption characteristics of silica toward removal of ionic organic dye such as methylene blue (MB) from water, with an adsorption capacity of 305 mg·g^−1^ [20]. It was also previously reported that the modification of silica with inorganic CuO or Fe improved their adsorption capacity to remove the cationic dyes, such as crystal violet and MB, due to the improvement of electrostatic attraction toward organic dyes and the enhancement of the adsorbent pores [21]. In our previous works, the surface of silica was modified with amphiphilic surfaces to increase their ability to adsorb the organic dye and heavy metal water pollutants [22]. In this respect, the present work aims to modify the porosity and surface activity of the silica nanoparticles by forming the smart surface active nanogels as a core or shell during the hydrolysis of silica using sol-gel technique. The smart nanogels are based on the crosslinking polymerization of *N*-isopropyl acrylamide (NIPAm) or 2-acrylamido-2-methylpropane sulfonic acid (AMPS) monomers with vinyltrimethoxysilane using surfactant-free method. The silicate/nanogel colloid is hydrolyzed with tetraethoxysilane (TEOS) to produce amphiphilic silica/nanogel nanocomposite. The surface of the incinerated porous silica nanoparticles was modified with amine functional group to apply as adsorbent for the removal of Coomassie blue (CB-R250). Coomassie brilliant blue (CBB), an anionic organic dye with two similar triphenylmethane, is synthesized to be used in the textile manufacture but now in protein determination and analytical biochemistry. It is a highly toxic dye and can lead to harmful effects carcinogenic consequences on the human body [23]. There are many developed methods to purify water from the CBB pollution. Some methods rely on the photocatalytic degradation of it [24]. One of the main problems of the photocatalytic degradation is the formation of secondary pollution. In our work, the dye was removed by adsorption mechanism with high-adsorption capacity and with no secondary pollutants formed.

## 2. Experimental

### 2.1. Materials

The chemicals were purchased from Sigma-Aldrich Co. (Missouri, MO, USA) and used as received. Acrylamide monomers based on *N*-isopropylacrylamide (NIPAm), 2-acrylamido-2-methylpropane sulfonic acid (AMPS), *N,N*-methylenebisacrylamide (MBA), and water-soluble radical initiator ammonium persulfate (APS) were used to prepare the crosslinked microgels. The vinyltrimethoxysilane (VTS) monomer is used as comonomer during the radical polymerization as silicone precursor to obtain microgels. Tetraethoxysilane (TEOS), 3-aminopropyltriethoxysilane (APTS), ammonium hydroxide solution (25 wt %), and toluene were used to hydrolyze the VTS.

### 2.2. Preparation Techniques

#### 2.2.1. Preparation of Silica Nanoparticles (Si NPs)

The half weight of NIPAm monomer (0.9 mol %; 1.2 g) dissolved in 100 mL of deionized water under nitrogen atmosphere was preheated at 40 °C for 30 min. The pre-polymerization of NIPAm started by injection of APS solution (0.08 g dissolved in 2 mL of the deionized water) followed by gradual ramp of the reaction temperature from 40 to 55 °C with a heating rate 0.33 °C·min^−1^. The remaining NIPAm (remained weight of 0.9 mol %; 1.2 g), VTS (0.1 mol %; 0.337 mL), and MBA (6 mol %; 0.06 g) were dissolved in 100 mL of deionized water under the nitrogen atmosphere and added to the reaction mixture with a feeding rate of 0.4 mL·min^−1^ using a syringe pump resulting in a turbid solution. The reaction temperature increased up to 65 °C for 4 h to obtain NIPAm (0.9)/VTS (0.1) microgel. The reaction was terminated by adding methanol, and the microgel was isolated and purified by ultracentrifugation of the reaction mixture at 8000 rpm. The microgel was washed several times with ethanol. The AMPS (0.05 and 0.1 mol %) was added to the NIPAm, VTS (0.05 mol %; 0.168 mL), and MBA mixture to prepare NIPAm (0.9)/AMPS (0.05)-VTS (0.05). NIPAm/VTS or NIPAm/AMPS-VTS microgel (2 g) was sonicated in 100 mL of deionized water to obtain microgel dispersion, followed by adding ammonia solution (28 wt %; 7 mL) to obtain clear dispersion. TEOS (8 mL) dispersed in n-hexane (20 mL) and injected dropwise under continuous stirring for 40 min to produce emulsion. The continuous stirring (200 rpm) continued for 24 h and the reaction temperature was around 35–40 °C. The hydrolyzed silica colloids were separated by ultracentrifugation of the mixture at 12,000× *g* rpm and washed with acetone and ethanol. The silica nanoparticles (Si NPs) were obtained from incineration of silica at 700 °C for 24 h.

#### 2.2.2. Preparation of SiO_2_ NPs

The silica nanoparticles (1 g) produced from incineration of NIPAm (0.9)/VTS (0.1) microgel colloid was suspended in dry toluene (50 mL) using sonication. APTS (1 mL) was added dropwise to the reaction mixture under continuous stirring. The reaction temperature was increased to the reflux temperature of 120 °C for 2 h. The ethanol and toluene was collected from the reaction mixture by using the vacuum rotary evaporator at 64 °C and 132 Torr for 90 min. The remaining solid was extracted using diethyl ether and dichloromethane (1:1 Volume %) by Soxhlet for 24 h. The SiO_2_ NPs produced from Si NPs based on NIPAm (0.9)/AMPS (0.05)-VTS (0.05) and NIPAm (0.8)/AMPS (0.1)-VTS (0.1) were designated as SiO_2_-5-5 and SiO_2_-10-10, respectively. The produced SiO_2_ NPs produced from Si NPs based on NIPAm (0.9)/VTS (0.1) and NIPAm (0.8)/VTS (0.2) were abbreviated as SiO_2_-10 and SiO_2_-20, respectively.

### 2.3. Characterization of the Prepared Composites

Fourier transform infrared analysis (FTIR; model Nexus 6700 FTIR; Nicolet Magna Newport, NJ, USA) helped to investigate the modification of the silica nanoparticles with amine groups. The thermal stability and organic contents on the silica nanoparticles were evaluated by thermogravimetric analysis (TGA; NETZSCH STA 449 C instrument, New Castle, DE, USA). The effect of pH aqueous solutions on the surface charges, hydrodynamic diameter, and polydispersity index (PDI) of amino-modified silica nanoparticles were investigated from dynamic light scattering (DLS) (Zetasizer Nano; Malvern Instrument Ltd, London, UK). The morphologies of amino-modified silica nanoparticles were examined by transmission electron microscopy (TEM; JEOL JEM-2100F; JEOL, Tokyo, Japan). The nitrogen adsorption desorption isotherms of the mesoporous amino-modified silica nanoparticles were evaluated using a Belsorp-mini II (BEL; Tokyo, Japan). The amino-modified silica nanoparticles sample was treated in vacuum oven at 100 °C under 10−30 mmHg before measurement. The calculated pore diameter was determined by using the Barrett-Joyner-Halenda (BJH) method. The calculated specific surface area of amino-modified silica nanoparticles sample was obtained using adsorption data in the *P*/*P*0 considering the linearity of a Brunauer-Emmett-Teller (BET) plot. UV-visible spectrophotometer (Shimadzu UV-1208 model; Kyoto, Japan) was used to determine the CB-R250 concentrations in water at 580 nm.

### 2.4. CB-R250 Adsorption Measurements

Adsorption characteristics of the amine SiO_2_-10, SiO_2_-5-5, SiO_2_-10-10, and SiO_2_-20 NPs adsorbents were carried out using batch modes. Stock aqueous solutions of the standard CB-R250 (from 0.01 to 0.1 mmol·L^−1^ using double distilled water) were used to demonstrate the calibration curve of CB-R250. The pH of the dye CB-R250 aqueous solutions adjusted using phosphate buffer solutions ranged from 2 to 10 with 0.01 M ionic strength. For each experiment, a known concentration of CB-R250 dye (15 mL) was taken and optimum quantities (mg) of SiO_2_-10, SiO_2_-5-5, SiO_2_-10-10, and SiO_2_-20 NPs adsorbents were added and kept at 25 ± 1 °C with agitation at a constant speed (150 rpm/min). The samples were then collected and centrifuged at 3000 rpm for 10 min. The CB-R250 absorbance determined by UV–visible spectrophotometer is used to determine the supernatant dye concentration at 580 nm. The maximum adsorption capacities at equilibrium *q_e_* (mg·g^−1^) and their removal adsorption efficiency (%) were calculated as:
*q*_max_ = (Co − Ce) × V/m(1)
E (%) = (Co − Ce) × 100/Co(2)

The Co, Ce, V, and m are the initial CB-R250 concentration in aqueous solutions, equilibrium CB-R250 concentration (mg·L^−1^), the volume of aqueous solution (L), and the adsorbent mass (g), respectively.

The desorption and reuse of SiO_2_-10, SiO_2_-5-5, SiO_2_-10-10, and SiO_2_-20 NPs adsorbents were obtained after the dispersion of CB-R250 saturated adsorbents in ethanol at pH 10 as an effective eluent and dried in vacuum oven. The measurements were repeated for five consecutive cycles.

## 3. Results and Discussion

The synthesis route of amine-modified SiO_2_ NPs reported in the experimental part is represented in Figure 1. In this respect, NIPAm monomer is selected as the temperature-sensitive monomer to be used in the formation of crosslinked nanogel, in the presence of MBA as a crosslinker, and to control their morphologies and particle sizes, which is attributed to the presence of hydrophilic amide and hydrophobic isopropyl group in its chemical structure [25,26,27]. It is proposed that the presence of NIPAm more than 80 mol % in the crosslinked polymers controlled the transformation of gels from microgels to nanogels [25,26,27]. Moreover, VTMS monomer is used as a source for the hydrolysis of the produced NIPAm nanogel resulting in the formation of silica colloids by using sol-gel technique when hydrolyzed with TEOS (Figure 1). The proposed VTMS is used to control the hydroxyl-group contents on the SiNPs surfaces. The AMPS hydrophilic anionic monomer is used to improve the hydrophilicity and morphology of the silica colloids after hydrolysis. In the first step, the monomers NIPAm, AMPS, and VTMS, with MBA as a crosslinker, were allowed to interact to result in the formation of the hydrogel incorporated with silicate as the organic core of the final silicate nanoparticle. Secondly, TEOS was added and basic hydrolysis had occurred using ammonia solution resulting in the formation of the outer silicate shell. After that, the formed silica composite was calcined at 700 °C to obtain mesoporous silica nanoparticles. Finally, APTS was utilized to functionalize the prepared mesoporous silica nanoparticle with amino group to make it pH-sensitive and to use it for Coomassie blue dye (CB-R250) removal from aqueous solutions.

### 3.1. Characterization of Amine Modified SiO_2_ NPs

The chemical structures of the amino functionalized SiO_2_-10, SiO_2_-5-5, SiO_2_-10-10, and SiO_2_-20 NPs were confirmed by FTIR spectra as summarized in Figure 1a–d. The appearance of bands in all spectra at 460–470 cm^−1^ (Si–O rocking vibration), 804–810 cm^−1^ (Si–O–H…H_2_O bending vibration), 1093–1107 cm^−1^ (Si–O–Si stretching vibration), 1635 cm^−1^ (OH bending vibration of the adsorbed water), and 3429–3464 cm^−1^ (OH stretching vibration, hydrogen bonded) elucidates the formation of SiO_2_ [28]. The appearance and increasing the intensity of bands at 3550–3590 cm^−1^ (N–H stretching), 385–3910 cm^−1^ (C–H aliphatic stretching), and 1550–1580 cm^−1^ (N–H bending) in the spectra of SiO_2_-5-5 and SiO_2_-10-10 (Figure 1b,c) more than that observed in FTIR spectra of SiO_2_-10 and SiO_2_-20 (Figure 1a,d) confirm the presence of more amino-propyl group at the SiO_2_ NPS surfaces.

The thermal stability and the contents of the isopropyl amine on the SiO_2_-10, SiO_2_-5-5, SiO_2_-10-10, and SiO_2_-20 NPs surfaces were determined from TGA thermograms represented in Figure 2a–d. The remained residual at temperature 650 °C in thermograms of NIPAm/VTS (Figure 2a,d) elucidates that the nanogel is completely degraded. The remained residue of NIPAm/VTS/AMPS nanogel in thermograms was approximately 10 wt % (Figure 2b,c) and confirmed that the presence of the AMPS in the crosslinked polymers produced more stable cyclic structures above 650 °C [29]. It was also noticed that the residual at 650 °C for the Si colloids of NIPAm(80)/VTS(20), NIPAm(80)/VTS(10)/AMPS(10), NIPAm(90)/VTS(5)/AMPS(5), and NIPAm(90)/VTS(10) are 50, 42, 28 and 34 wt %, respectively (Figure 2a–d). These data elucidate the formation of si colloid with the increase of VTS content and incorporation of AMPS. By comparing the remained residual of Si NPs after calcination and the amine-modified SiO_2_-10, SiO_2_-5-5, SiO_2_-10-10, and SiO_2_-20 NPs, it is found that the percentages of amine modifications are 12, 17, 25, and 10 wt %, respectively. This means that the modification of Si NPs surfaces ordered as SiO_2_-10-10 > SiO_2_-5-5 > SiO_2_-10 > SiO_2_-20 confirms that the surfaces of the produced Si NPs easily hydrolyzed with APTS with the increase in both VTS and AMPS contents. The TGA thermograms (Figure 2a–d) confirm that the thermal stability of nanogels improved by converting them to Si colloids and Si or modified SiO_2_ NPs.

The morphology of the SiO_2_-5-5, SiO_2_-10, SiO_2_-10-10, and SiO_2_-20 NPs surfaces determined by TEM micrographs are shown in Figure 3a–d. The uniform core/shell morphology appeared in SiO_2_-5-5 and SiO_2_-10-10 micrographs (Figure 3a,c) more than SiO_2_-10 and SiO_2_-20 (Figure 3b,d). Moreover, the particle sizes of the dry SiO_2_-5-5 and SiO_2_-10-10 are 10–15 nm and 45–55 nm, respectively, while the dry SiO_2_-10 and SiO_2_-20 NPs tend to form aggregates. It was also noticed that the morphology of the amine modified SiO_2_-5-5 NPs (Figure 3a) have core and shell thicknesses 10–12.5 and 7.5–10 nm that appeared as dark core and bright shell, respectively. The SiO_2_-10-10 NPs micrograph (Figure 3c) shows core and shell morphology with thicknesses 35–45 and 10–15 nm, respectively. This means that the thickness of amine shell increased for the SiO_2_ NPs with increasing VTS contents as proved from FTIR spectra (Figure 1a–d). The increase in the hydroxyl-group contents in the Si NPs formed after calcination tends to increase the hydrolysis with the ethoxy groups of APTS, forming bright shells, and functionalize the silica surfaces with –NH_2_ and –OH groups. The lowering of –NH_2_ and –OH groups on the surfaces of SiO_2_-10 and SiO_2_-20 (Figure 3b,d) tends to form aggregates due to the formation of intermolecular hydrogen bonds among their particles [30].

It is very important to determine the surface area (m^2^·g^−1^), pore volume (cm^3^·g^−1^), and pore size diameter (nm) of the modified SiO_2_ NPs to elucidate their porosity and adsorption characteristics. These parameters were measured from N_2_ adsorption-desorption isotherms using BET plots as summarized in Figure 4a–d and Table 1. The adsorption-desorption isotherms of all samples show the classical type IV isotherms except SiO_2_-20 (Figure 4a). The classical type IV adsorption-desorption isotherms (Figure 4b–d) show mesoporosity without hysteresis between the adsorption and desorption with mesoporous sizes about 2 nm. Whereas, the pore size distribution of SiO_2_-10-10 (Figure 4b) and SiO_2_-5-5 (Figure 4c) show sharp and narrow distribution at 2.12 and 1.5 nm. The SiO_2_-20 adsorption-desorption (Figure 4a) shows much longer straight line portion of the curve which is its starting point known as the adsorption capacity of monolayer, as defined by Brunauer and Emmett [31]. However, multilayers of the adsorbate were formed on the surfaces with the gradual increase in pressure. It was noticed that from the data listed in Table 1 that the pore sizes decrease in the order SiO_2_-20 > SiO_2_-10 > SiO_2_-5-5 > SiO_2_-10-10 to elucidate that the grafting of the pores of SiO_2_ with APTS decreases with the increase in the amine group contents.

### 3.2. Effect of pH on the Surface and Particle Sizes of SiO_2_ NPs

The surface charges of the amine-functionalized SiO_2_ NPs were estimated from their zeta potential measurements in aqueous solution at discriminate pH values between 2 and 12 as represented in Figure 5. The isoelectric points of SiO_2_-10, SiO_2_-20, and SiO_2_-5-5 NPs determined as 6.3, 7.3, and 5.0, which were attributed to pH, produced zero surface charge, and elucidated that only pH-sensitive SiO_2_-10, SiO_2_-20, and SiO_2_-5-5 NPs were prepared. The presence of the functional groups on the SiO_2_ NPs influenced their zeta potentials. All amine-functionalized SiO_2_ NPs have positive zeta potentials on their surfaces in acidic pH aqueous solution that referred to protonation of amine functional groups. The negative zeta potentials of amine-functionalized SiO_2_ NPs in alkaline aqueous solution can be attributed to the presence of silanol hydroxyl groups that converted to negative charge at pH > 7 and lone pairs of amine groups. The negative surface charges of the SiO_2_ NPs produced at slightly acidic pH from 5.6 to 6.9 and higher NIPAm contents >80 mol % confirm the higher porosity and more amine groups at the SiO_2_ NPs surfaces. The positive surface charges of SiO_2_ NPs produced from gels having NIPAm contents 80 mol % at pH 7–8 elucidate the acidity of the amine-functionalized SiO_2_ NPs surfaces after calcination. The acidity of SiO_2_ NPs surfaces increased more with the incorporation of AMPS (10 mol %), which shows positive surface charges even at pH 10. One of the most generally accepted explanation for the basicity and positive charges of calcinated nanomaterials is the formation of electron-donor-acceptor complex [32,33]. It was reported that the oxygen-free carbon sites on the nanomaterials can easily absorb oxonium ions H_3_O^+^ to provide positive and hence basic properties [34]. Therefore, although the SiO_2_ NPs surfaces have been functionalized with reactive amine group, the positive surface charges created on their surface after calcination due to absorbed oxonium ions H_3_O^+^ contribute more significantly to the surface potential of the particles. The ability to reverse their surface charges from positive to negative with pH variation increased in the order SiO_2_-5-5 > SiO_2_-10 > SiO_2_-20 and was omitted for SiO_2_-10-10 NPs.

The effect of pH of aqueous solution on the particle sizes of SiO_2_-10, SiO_2_-5-5, SiO_2_-20, and SiO_2_-10-10 NPs is investigated by using DLS measurements as represented in Figure 6a–d. The polydispersity index (PDI) and particle-size diameters of the amine-modified SiO_2_ NPs are determined and listed in Figure 6a–d. All amine-modified SiO_2_ NPs formed aggregates or cluster at pH 7 and their sizes were altered by changing the pH of aqueous solutions either in acidic or alkaline aqueous solutions (Figure 6a–d). The amine-modified silica nanomaterials tend to form interparticle network and consequently gelation, which increased with increasing the amine contents [35,36,37,38,39]. The increasing of the particle sizes of SiO_2_-10-10 NPs (Figure 5d) more than other amine-modified SiO_2_ NPs agrees with the previous works [35,36,37,38,39] and elucidates the increasing of amine contents on its surfaces as confirmed from FTIR and TGA analysis (Figure 1a–d and Figure 2a–d, respectively). The high positive charge density on the surfaces of SiO_2_-10-10 NPs as elucidated from zeta potentials at different pHs (Figure 5) can also induce gelation of silica [40]. The lower amine contents of SiO_2_-10 and SiO_2_-20 (Figure 6a,c) reduce the particle sizes of aggregates at different pH values. These data confirm that the amine-modified SiO_2_ NPs have pH sensitivity as confirmed from both surface charges and particle-size measurements to elucidate their applicability for medical and environmental applications.

### 3.3. Sequestration and Optimization of CB-R250 Dye Adsorption

The surface charges of the prepared SiO_2_-10, SiO_2_-5-5, SiO_2_-20, and SiO_2_-10-10 NPs are positively charged in aqueous acidic pH 4 and negatively charged in alkaline solution above pH 8 except SiO_2_-10-10 (Figure 5). Moreover, all SiO_2_-10, SiO_2_-5-5, SiO_2_-20, and SiO_2_-10-10 NPs are aggregated at pH 7 and well dispersed either in acidic or alkaline solution (Figure 6a–d) to confirm their sensitivity to pH of the solution. The nitrogen adsorption-desorption data (Figure 4a–d) elucidate the formation of mesoporous adsorbents having higher surface area and lower pore volume and pore size diameter except SiO_2_-20 due to lower contents of amine groups at its surface. These data confirm the excellent adsorption characteristics of the prepared SiO_2_ NPs, which can be used as nano-adsorbent for organic dye water pollutants. In this respect, CB-R250 is one of the non-biodegradable anionic toxic reactive dyes that cannot be easily removed by several adsorbents having adsorption capacities more than 120 mg·g^−1^ [41,42,43,44,45,46]. The goal of the present work is using mesoporous SiO_2_-10, SiO_2_-5-5, and SiO_2_-10-10 NPs under optimization conditions for nano-adsorbents to remove CB-R250 dye from aqueous solution. The optimum conditions such as adsorbents weights, CB-R250 concentration, and pH of aqueous solution have been determined to effectively remove CB-R250 pollutant. The prepared Si NPs cannot be used to remove CB-R250.

The optimum SiO_2_-10, SiO_2_-5-5, and SiO_2_-10-10 NPs adsorbents’ weight to remove 100 mg·g^−1^ of CB-R250 dye (0.06 mmol·L^−1^) are determined from the relation of dye removal efficiencies (%) and their weights (mg) as plotted in Figure 7. Their values are 6, 6, and 7.5 mg for SiO_2_-5-5, SiO_2_-10-10, and SiO_2_-10, respectively. The data confirm that both SiO_2_-5-5 and SiO_2_-10-10 effectively remove the CB-R250 dye more than SiO_2_-10 due to an increase in the amine contents on their surfaces that are easily protonated to produce positively charged adsorbents more than SiO_2_-10. These data agree with the FTIR spectra (Figure 1a–d) and TGA thermograms (Figure 2a–d) that confirmed the increase of amine modification on the SiO_2_-5-5 and SiO_2_-10-10 surfaces increases their ionic interactions with negatively charged CB-R250 dye.

The optimum initial CB-R250 adsorbates’ concentrations using 6, 6, and 7.5 mg of SiO_2_-5-5, SiO_2_-10-10, and SiO_2_-10 adsorbents are determined from Figure 8. The removal efficiency values elucidate that the SiO_2_-10-10 adsorbed 100% at initial concentrations of 100 (0.12 mmol·L^−1^) and 50 mg·L^−1^ (0.06 mmol·L^−1^) of CB-R250. The removal efficiency values of SiO_2_-5-5 and SiO_2_-10 adsorbents decreased progressively with increasing the initial CB-R250 concentration more than 50 mg·L^−1^ (0.06 mmol·L^−1^). The removal efficiency values of SiO_2_-10 reincreased to 95% at initial CB-R250 concentration 200 mg·L^−1^ (0.12 mmol·L^−1^). The higher reactive amine group sites of SiO_2_-5-5 and SiO_2_-10-10 increase the concentration of the adsorbed CB-R250 on their surfaces at lower CBB concentration [47]. While the lower pore sizes and pore volume of SiO_2_-10 (Table 1 and Figure 4a–d) increase the CB-R250 adsorption at a higher concentration 200 mg·L^−1^ (0.12 mmol·L^−1^).

The pH plays an important role for the removal of the charged reactive dyes from their aqueous solutions using adsorbents. The effect of pH on the removal efficiency of optimum weights of SiO_2_-5-5, SiO_2_-10-10, and SiO_2_-10 adsorbents to remove the optimum initial CB-R250 concentration at 50 mg·L^−1^ (0.06 mmol·L^−1^) is represented in Figure 9. It is well known that the CB-R250 has zwitter ionic property due to the presence of amine cations and sulfonate group on its surface, and the change of pH results in the formation of different ionic species and different SiO_2_ NPs charges [48]. It was observed that (Figure 9) that SiO_2_-10 adsorbent shows maximum adsorption at pH 4 and 7. The SiO_2_-5-5 adsorbent achieved maximum adsorption at pH 4, and its adsorption decreased with increasing pH. The SiO_2_-10-10 adsorbent achieved good CB-R250 removal in acidic and alkaline pH values, and the optimum removal occurred at pH 4. These data can be correlated with the DLS and surface charges data (Figure 5 and Figure 6) that elucidate that the SiO_2_-5-5, SiO_2_-10-10, and SiO_2_-10 aggregate at pH 7 and dispersed in either acidic or basic aqueous solutions. The positive charges of SiO_2_-10-10 in both acidic and alkaline pH (Figure 5) increase its use in the removal of the zwitter ionic CB-R250 either in acidic or basic solutions.

### 3.4. Adsorption Isotherms

The surface homogeneity/heterogeneity of the adsorbents and the assembly of the adsorbate on the adsorbent surface via formation of either monolayer or multilayers can be evaluated by applying the most applicable Langmuir and Freundlich isotherms. The optimum adsorption conditions reported in the previous section used to fulfil the Langmuir and Freundlich isotherms from the Equations:(*C*_e_/*q*_e_) = [(1/*Q*_max_*K*_l_) + (*C*_e_/*Q*_max_)] (3)
log(*q*_e_) = log(*K*_f_) + [(1/*n*) log(*C*_e_)] (4)

The *C*_e_ (mg·L^−1^) is the concentration of CB-R250 dye in the aqueous solution at equilibrium. The *n* (in g·L^−1^), *K*_l_ (in L·mg^−1^), and *K*_f_ (in (mg·g^−1^)(L·mg^−1^)^(1/*n*)^) constants reported in Equations (3) and (4) designated the empirical constant, Langmuir constant, and Freundlich constant, respectively. The maximum experimental adsorption capacities (*q*_e_; mg·g^−1^) and the theoretical value of the adsorbents (*Q*_max_; mg·g^−1^) are determined and listed in Table 2. Equations (3) and (4) should obey a linear relation with the highest linear coefficient (*R*^2^). The Langmuir and Freundlich parameters were calculated from the plots that obeyed Equations (3) and (4) and summarized in Table 2. The data, listed in Table 2, prove that all SiO_2_-5-5, SiO_2_-10-10, and SiO_2_-10 adsorbents fit the linear Langmuir model due to the agreement of the theoretical *Q*_max_ with the experimental *q*_e_ and the highest *R*^2^ values. These data elucidate that the SiO_2_-5-5, SiO_2_-10-10, and SiO_2_-10 have homogeneous surfaces and the CB-R250 dye forms monolayer on their surfaces.

The presence of amine groups on the adsorbent surfaces of SiO_2_-5-5, SiO_2_-10-10, and SiO_2_-10 adsorbents proposed that the alkaline ethanol is estimated as the best reagent for their desorption data and regeneration [49]. In this respect, ethanol at pH 10 is used to desorb the CB-R250 from the prepared SiO_2_-5-5, SiO_2_-10-10, and SiO_2_-10 adsorbents. The relation of *q_e_* data versus six cycles were summarized in Figure 10. It can be seen that both the SiO_2_-5-5 and SiO_2_-10-10 easily desorbed and reused with little changes of *q_e_* more than SiO_2_-10. These data can be referred to the higher amine contents and porosity of both SiO_2_-5-5 and SiO_2_-10-10 more than SiO_2_-10 adsorbents to enhance the chemical bonding of CB-R250. Finally, it can be concluded that the prepared SiO_2_-5-5, SiO_2_-10-10, and SiO_2_-10 were adsorbed and reused as effective adsorbents for CB-R250 than the reported data in the literature [41,42,43,44,45,46].

## 4. Conclusions

New amino-functionalized smart nanoparticles were prepared by applying sol-gel technique for hydrolyzing NIPAm/VTS and NIPAm/VTS/AMPS microgels with TEOS and APTS. The FTIR and TGA data elucidated that the aminomodification of Si NPs surfaces ordered as SiO_2_-10-10 > SiO_2_-5-5 > SiO_2_-10 > SiO_2_-20 to confirm that the surfaces of the produced Si NPs easily hydrolyzed with APTS with increasing VTS and AMPS contents. The uniform core/shell morphology appeared in the presence of AMPS in both SiO_2_-5-5 and SiO_2_-10-10 micrographs more than SiO_2_-10 and SiO_2_-20. The nitrogen adsorption-desorption isotherms show the mesoporosity of amino-functionalized silica NPs without hysteresis between the adsorption and desorption with mesoporous sizes about 2 nm. The pore sizes decrease in the order SiO_2_-20 > SiO_2_-10 > SiO_2_-5-5 > SiO_2_-10-10 to elucidate the grafting of the pores of SiO_2_ with APTS to decrease with increasing amine group contents. The surface charges of the prepared SiO_2_-10, SiO_2_-5-5, SiO_2_-20, and SiO_2_-10-10 NPs are positively charged in aqueous acidic pH 4 and negatively charged in alkaline solution above pH 8 except SiO_2_-10-10. Moreover, all SiO_2_-10, SiO_2_-5-5, SiO_2_-20, and SiO_2_-10-10 NPs are aggregated at pH 7 and well dispersed either in acidic or alkaline solution to confirm their sensitivity to pH of the solution. The adsorption removal capacities of the prepared amino-functionalized silica nanoparticles toward CB-R250 increased with increasing their amino contents on their surfaces. Moreover, the mesoporous structure and higher surface area are showing favorable adsorptive removal of CB-R250 with higher experimental adsorption capacity of 313 mg·g^−^^1^, which was not reported previously in the literature. From the view of environmental impact, amino-functionalized smart silica NPs could be regarded as effective adsorbent in future water treatment.

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
