# Peer review of "Reactive Mesoporous pH-Sensitive Amino-Functionalized Silica Nanoparticles for Efficient Removal of Coomassie Blue Dye"

_nanomaterials, 2019, doi:10.3390/nano9121721_

Round 1

Reviewer 1 Report

The authors report mesoporous amine-functionalized silica nanoparticles from hydrolyzing microgels based on N-isopropyl acrylamide -co- vinyltrimethoxysilane  microgels with tetraethoxysilicate and 3-aminopropyltriethoxysilane by sol-gel method. Although the detail of materials is not same and the referee could find some novelty, the materials can be included in a large framework of amine functionalized porous silica. Compare to previously reported materials, the referee could not find good enough novelty to be accepted in nanomaterials in current form. And applications are also at a basic level. Thus, the referee think that current version of manuscript is not acceptable to be published in nanomaterials.

Each step shown in the scheme needs to well analyze. In figure 3, magnification of each figures are different. a) Images of the same scale need to be included. b) An additional high magnification TEM images for each materials should also be included to show the pore. Compare the previously reported amino-functionalized mesoporous and emphasize that what advantages your material have and how?? If detection of CB-R250 is important issue in pollution field, the authors need to mention it. Otherwise, additional data for separation of an environmental pollutant that is in desperate need of separation can be helpful. Figure 5, the x-axis writing is broken or overlaid.

Reviewer 2 Report

Following an overall inquiry into the reviewed article, I consider it to be a very interesting investigation of the mesoporous amino-functionalized silica nanoparticles for removal of Coomassie blue dye. The manuscript contains interesting and valuable data, which have been mostly correctly evaluated and interpreted. Organization and clarity of the manuscript is also generally good. The paper resolves an elaborate topic and meets formal layout standards and default criteria, imposed on such articles. Thereby, I commend its issuance. I would like to ask the authors to marginally mention the following article in the introduction:

Galamboš M., Suchánek P., Rosskopfová O.: Sorption of anthropogenic radionuclides on natural and synthetic inorganic sorbents. J. Radioanal. Nucl. Chem. 293(2): 613-633 (2012).

Reviewer 3 Report

This manuscript clearly describes the amino-functionalized mesoporous silica nano gel preparation which is pH sensitive material that can solve the problem of limitation of silica nano gel limitation in application organic pollutants removal. However, the authors fail to highlights those above things discussion in their background works compared to other similar types of materials at the introduction. After implementing the following comments, it can be accepted for publication.

The authors should be the highlight this study's significance and novelty with compared the similar type of mesoporous silica hydrogel nanoparticles [ For examples: Nanomaterials (Basel). 2018 Jan; 8(1): 4. Published online 2017 Dec 23. doi: 10.3390/nano8010004; Process Safety and Environmental Protection 109, DOI: 10.1016/j.psep.2017.04.011 etc.] Maintain the consistency and standards in the representation of units thought the text, Examples: Line 95, 'ml' should be 'mL' and Line 81, "4 hrs" should be "4 h" Line 92 and 99, "24 hrs" should be "24 h"  'minutes' should be 'min' and 'M' should be 'mol L-1' I felt that the first para of results discussion is the background of this proposed work. It should be kept at introduction. Line 160, 'minofunctionalized.......' should be correct. Fig.5 scale not clear appearance, re-draw it. Add error bars for all adsorption data to understand the reproducibility and precision of measurements.

Round 2

Reviewer 1 Report

Although some of issues were solved, still the referee think that the material and application is not good enough to be published in nanomaterials which is high IF journal.

1) If the material is innovative materials, the authors need to mention it clearly. The material may have a little better properties, but there is nothing surprising compared to the existing material.

2) Quality of presentation (especially figures) is very poor. a) Still scale bar is different scale and shape in Figure 3. b) Most of figures are poor in terms of quality, layout and arrangement. c) Miswording. “Figure 64” must be miswording. Ex) Figure 64a-d and Table 1. The adsorption-desorption isotherms of all samples show 220 the classical type IV isotherms except SiO2-20 (Figure 64a).